# Autophagy Inhibits Intercellular Transport of Citrus Leaf Blotch Virus by Targeting Viral Movement Protein

**DOI:** 10.3390/v13112189

**Published:** 2021-10-30

**Authors:** Erbo Niu, Huan Liu, Hongsheng Zhou, Lian Luo, Yunfeng Wu, Ida Bagus Andika, Liying Sun

**Affiliations:** 1State Key Laboratory of Crop Stress Biology for Arid Areas, College of Plant Protection, Northwest A&F University, Xianyang 712100, China; neb1991@163.com (E.N.); lianziqian0709@163.com (H.Z.); LiianLuo@163.com (L.L.); 2School of Modern Agriculture and Biotechnology, Ankang University, Ankang 725000, China; liuh2020@126.com; 3College of Plant Health and Medicine, Qingdao Agricultural University, Qingdao 266109, China

**Keywords:** autophagy, antiviral machinery, citrus leaf blotch virus, protein degradation, virus movement

## Abstract

Autophagy is an evolutionarily conserved cellular-degradation mechanism implicated in antiviral defense in plants. Studies have shown that autophagy suppresses virus accumulation in cells; however, it has not been reported to specifically inhibit viral spread in plants. This study demonstrated that infection with citrus leaf blotch virus (CLBV; genus *Citrivirus*, family *Betaflexiviridae*) activated autophagy in *Nicotiana benthamiana* plants as indicated by the increase of autophagosome formation. Impairment of autophagy through silencing of *N. benthamiana* autophagy-related gene 5 (*NbATG5*) and *NbATG7* enhanced cell-to-cell and systemic movement of CLBV; however, it did not affect CLBV accumulation when the systemic infection had been fully established. Treatment using an autophagy inhibitor or silencing of *NbATG5* and *NbATG7* revealed that transiently expressed movement protein (MP), but not coat protein, of CLBV was targeted by selective autophagy for degradation. Moreover, we identified that CLBV MP directly interacted with NbATG8C1 and NbATG8i, the isoforms of autophagy-related protein 8 (ATG8), which are key factors that usually bind cargo receptors for selective autophagy. Our results present a novel example in which autophagy specifically targets a viral MP to limit the intercellular spread of the virus in plants.

## 1. Introduction

Autophagy is a highly conserved process that degrades and recycles any unnecessary or damaged cytoplasmic components [1,2]. Basal levels of autophagy maintain cellular homeostasis under normal environmental conditions, while higher levels of autophagy are induced by various abiotic and biotic stress cues including nutrient deficiency, and pathogen infection [3,4]. There are three major types of autophagy in plants: micro-autophagy, macro-autophagy, and mega-autophagy [5,6]. Among them, macro-autophagy is considered to be the major process for degrading cytoplasmic proteins and organelles, and hereafter it is referred to as autophagy [1,7]. Autophagy is initiated by the formation of double-membrane vesicles, called autophagosomes, followed by fusing with lysosomes (in mammals) or vacuoles (in yeast and plants) to degrade and break down the enclosed cargoes [8].

The autophagy process requires a conserved set of proteins encoded by autophagy-related genes (ATGs) [9]. Among them, ATG8, a ubiquitin-like conjugation protein with a unique amino (*N*)-terminal extension, is essential for autophagosome formation and the recruitment of specific cargo receptors [10,11]. ATG8 proteins are covalently attached to the membrane lipid phosphatidylethanolamine (PE) by the ubiquitin-like conjugation system, which is essential for autophagosome formation and regulation of ATG8 function [12,13,14]. Although early-diverged plant lineages and various other eukaryotes carry only one ATG8 gene, higher plants have evolved multiple ATG8 isoforms with diverse and flexible functions [11]. In selective autophagy, ATG8-family proteins bind to cargo receptors or substrates via a conserved motif called ATG8-interacting motif (AIM) (or LIR-LC3II interacting region in mammals) and recruit them for the autophagic degradation [15]. The core motif of AIM can be written as W/F/Y-XX-L/I/V, which is composed of an aromatic amino acid W/F/Y, two amino acids XX, and a hydrophobic amino acid L/I/V [16]. Previous studies reported that autophagy could target plant viral proteins through direct recognition by ATG8 [17,18]. The various ubiquitin-like ligases, such as E1-like ligase ATG7, the E2-ligase ATG3, and the E3-like ligase ATG5, are considered to be critical components for the regulation of the autophagy process [1,9].

Several studies have revealed that autophagy is activated in response to various plant viral infections to limit virus accumulation, suggesting that the autophagic mechanism is evolutionarily conserved as a basal antiviral defense in plants; however, autophagy has also been shown to be involved in the promotion of viral infections [19,20,21,22]. Furthermore, viruses have evolved diverse measures to counteract autophagy for their own advantages [21,23,24,25,26,27]. Several reports have shown that autophagy restricts virus infection through selective degradation of proteins encoded by RNA viruses such as 2b, an RNA-silencing suppressor of cucumber mosaic virus (CMV) [28], RNA-dependent RNA polymerase of turnip mosaic virus (TuMV) [29], p3, an RNA-silencing suppressor of rice stripe virus (RSV) [30], the helper-component proteinase (HC-Pro), an RNA-silencing suppressor of TuMV, [31] as well as proteins encoded by double-stranded DNA (dsDNA) and single-stranded DNA (ssDNA) viruses [18,32,33].

Citrus leaf blotch virus (CLBV), a member of the genus *Citrivirus*, family *Betaflexiviridae* [34], naturally infects a wide range of plant species such as citrus, lemon, sweet cherry, peony, and kiwi [35,36,37,38]; depending on the plant hosts, it induces varied symptoms such as vein clearing, chlorotic blotching, and stem pitting [39,40]. Previously, CLBV was reported to be able to infect some *Nicotiana* species [41,42]. The CLVB genome is a single-stranded, positive-sense RNA [(+)ssRNA] consisting of 8747 nucleotides, excluding a 3′-terminal poly(A) tail, which encodes three open reading frames (ORFs) and is enclosed in filamentous and flexuous virions approximately 960 × 14 nm in size [43]. CLBV ORF1 encodes a ~227 kDa polyprotein thought to be a replication protein (replicase) containing methyltransferase, AlkB-like, Otu-like peptidase, papain-like protease, helicase, and RNA-dependent RNA polymerase (RdRp) motifs. ORF2 encodes a ~40 kDa protein with a sequence characteristic of viral movement protein (MP) of the 30 K superfamily, while ORF3 encodes a 41 kDa coat protein (CP) [44,45], and both proteins are expressed through subgenomic RNA (sgRNA) transcriptions (MP and CP sgRNAs) [44,45]. CLBV MP was shown to have an RNA silencing suppression activity [46].

The role of autophagy in modulating CLBV infection remains unknown. In this study, using the model plant *Nicotiana benthamiana*, we investigated whether autophagy operates to restrict CLBV infection. Overall, we observed that autophagy affected CLBV infection and viral protein accumulation in *N. benthamiana*.

## 2. Materials and Methods

### 2.1. Plant and Virus Materials

*N. benthamiana* plants were soil grown in environmental chambers under a 16-h light/8-h dark cycle at 25 °C. The full-length infectious clone of CLBV was kindly provided by Dr. Yunfeng Wu (Northwest A&F University, China). *Actinidia chinensis* (kiwifruit) plants infected with CLBV (GenBank accession no. MH427033) were obtained from a field in Zhouzhi County, in the Shaanxi Province of China, and used as a virus source.

### 2.2. Plasmid Construction

For the generation of the CLBV infectious clone, total RNA extracted from CLBV-infected leaves was subjected to cDNA synthesis using ReverTra Ace reverse transcriptase (Toyobo, Osaka, Osaka Prefecture, Japan). The full-length genome of CLBV containing poly(A) was amplified using the primers F-pCass-CLBV and R-CLBV. Subsequently, the hepatitis delta virus ribozyme sequence was inserted downstream of the 3′-UTR of the viral genome by PCR using the primers F-pCass-CLBV and R-pCass-RZ. PCR amplification and DNA ligation were performed using PrimeSTAR^®^ HS DNA Polymerase (Takara Bio, Kyoto, Japan) and a one-step cloning kit (Vazyme Biotech, Nanjing, China), respectively. PCR products were ligated between the *Kpn*I and *Sac*I sites of pCass4 [47] to produce the pCass-CLBV. A recombinant CLBV infectious clone carrying the green fluorescent protein (GFP) gene was generated as described previously with minor modifications [48]. The CLBV clone was digested with the restriction enzymes *Sal*I and *Pme*I and then a DNA fragment containing a partial MP coding sequence, the full length of the CP gene, a duplicate of the CP subgenomic RNA (sgRNA) promoter and the GFP gene, generated using overlapping PCR, was inserted into the same sites of the linearized infectious clone to produce pCass-CLBV-GFP. All of the primers used in this study are listed in Appendix A.

Replicase, CP, and MP ORFs were amplified from the CLBV infectious clone. NbATG8c1 (accession no. MG733101), NbATG8d (KX369400), NbATG8f (KU561372), and NbATG8i (KX369401)—coding sequences were obtained using RT-PCR with primers designed according to the sequence deposited in the National Centre for Biotechnology Information (NCBI) database.

For the transient expression assay, the full-length coding sequences of replicase, CP, MP, NbATG8c1, NbATG8d, NbATG8f, and NbATG8i were amplified by PCR, cloned into pBin41 [49], pBin61-GFP [50], pCambia1302-GFP [51] or pBI121-mCherry [52] digested with *BamH*I, *BamH*I, *Spe*I or *BamH*I, respectively, to generate CP-HA, MP-HA, GFP-ATG8c1, GFP-ATG8C1, GFP-ATG8d, GFP-ATG8f, GFP-ATG8i, replicase-GFP, CP-GFP, and MP-GFP, CP-mCherry, and MP-mCherry.

For the bimolecular fluorescence complementation (BiFC) assay, the coding regions of CLBV-MP (including MP mutant derivatives) and NbATG8 isoforms were inserted into the pBin61-CYFP or pBin61-NYFP [53] after digestion with the *BamH*I and *Sma*I sites to form CLBV-YFP(c)-MP, YFP(n)-NbATG8c1, YFP(n)-NbATG8C1, YFP(n)-NbATG8d, YFP(n)-NbATG8f, and YFP(n)-NbATG8i, respectively. The MP-N (nucleotides 1–543), MP-C (nucleotides 545–1089), MP-N84 (nucleotides 1–252), and MP-N90 (nucleotides 1–270) fragments were inserted into the BiFC vectors as described for full-length MP.

For MBP-tagged protein expression, the coding sequences of MP, MP-N, MP-C, and MBP-CP were inserted into the pMAL-c2X vector [54] between the *Xba*I and *Hind*III sites to generate MBP-MP, MBP-MP-N, MBP-MP-C, and MBP-CP, respectively. For tobacco rattle virus (TRV)-based virus-induced gene silencing (VIGS), a partial fragment of GUS (nucleotides 1059–1355; S69414), NbATG5 (nucleotides 1–300; KX369397), or NbATG7 (nucleotides 1–300; KX369398) was generated by PCR and then cloned into the pTRV2 vector [55].

### 2.3. Agroinfiltration

*Agrobacterium tumefaciens* (strain GV3101) cultures harboring binary vector constructs were resuspended in infiltration buffer (10 mM MES, pH 5.7, 10 mM MgCl_2_, and 150 mM acetosyringone). After 4 h incubation at room temperature, *Agrobacterium* cultures were infiltrated into *N. benthamiana* leaves. For inoculation of CLBV or CLBV-GFP, an *Agrobacterium* culture harboring RNA-silencing suppressor protein p19 [56] was added into the cultures.

### 2.4. Fluorescence Protein Observation

An Olympus FV3000 confocal laser scanning microscope was used to visualize GFP (excitation, 488 nm; emission, 510–550 nm), yellow fluorescent protein (YFP) (excitation, 514 nm; emission, 565–585 nm), mCherry (excitation, 543 nm; emission, 560–630 nm) and chlorophyll auto-fluorescence (excitation, 405 nm; emission, 635–708 nm) signals.

### 2.5. Maltose-Binding Protein (MBP) Pull-Down Assay

MBP-MP, MBP-MP-N, and MBP-MP-C fusion proteins were expressed in *Escherichia coli* (strain BL21) and purified as described previously [53]. Total proteins extracted from the GFP-NbATG8 isoform-expressing *N. benthamiana* leaves were incubated with purified MBP-MP, MBP-MP-N, or MBP-MP-C proteins bound to amylose resin (New England Biolabs, Ipswich, MA, USA). The proteins were washed with column buffer (20 mM Tris-HCl pH 7.4, 200 mM NaCl, and 1 mM EDTA pH 8.0) and boiled for 10 min. The interacting proteins were analyzed using an immunoblot with anti-GFP antibody.

### 2.6. Chemical Treatments

For the inhibitor assay, *N. benthamiana* leaves were pressure infiltrated with 100 µM E64d, 10 mM 3-MA, or 100 µM MG132 (Sigma-Aldrich, Saint Louis, MO, USA). Dimethyl sulfoxide (DMSO) was used as a negative control. For observing GFP-ATG8f-labeled autophagosomes, *N. benthamiana* leaves were vacuum-infiltrated with 100 µM E64d and kept for 8 h in the dark before observation.

### 2.7. Transmission Electron Microscopy (TEM) Observation

*N. benthamiana* leaves were cut into small fragments (1 × 4 mm) and infiltrated in 100 mM phosphate buffer (pH 7.0) containing 2.5% glutaraldehyde and 1% osmium tetroxide (OsO_4_). After post-fixation in OsO_4_, the samples were dehydrated in ethanol and then embedded in Epon 812 resin. The sections were cut from the embedded tissues on an ultramicrotome, and then stained with uranyl acetate and lead citrate before examination under a transmission electron microscope (Hitachi JEM-1230).

### 2.8. Immunoblot Analysis

Immunoblotting was performed as previously described [57]. The antibodies used in this study were as follows: primary anti-GFP (1:5000; Signalway Antibody Co., Ltd., College Park, MD, USA), anti-HA (1:2000; Abcam), anti-actin (1:5000; Kangwei, Taizhou, Jiangsu, China), and secondary goat anti-mouse IgG-HRP (1:5000; Proteintech, Wuhan, Hubei, China). For detection of CLBV CP protein, prokaryotically expressed recombinant MBP-CP was purified using amylose resin and then used to immunize rabbits. The obtained antiserum was used as the CP primary antibody (1:2000). Goat anti-rabbit IgG-HRP (1:5000; Proteintech) was used as the secondary antibody. For detection of ATG8-PE, total protein extracts from *N. benthamiana* leaves were separated on 15% sodium dodecyl sulphate–polyacrylamide gel electrophoresis (SDS-PAGE) with 6 M urea, followed by immunoblotting using an anti-ATG8 primary antibody (1:2000; Abcam) [58].

### 2.9. RT-PCR, Quantitative RT-PCR (qRT-PCR) and Northern Blot Analyses

Total RNA was extracted from plant tissues using Trizol (Invitrogen, Waltham, MA, USA) and first-strand cDNA was synthesized with ReverTra Ace reverse transcriptase (Toyobo, Osaka, Osaka Prefecture, Japan). For RT-PCR, gene fragments were amplified with specific primers (Appendix A) using 2× mixture DNA polymerase (Kangwei, Taizhou, Jiangsu, China). For qRT-PCR, PCR reactions were performed using the GoTaq^®^ Green Master Mix kit (Promega, Madison, WI, USA), and the 18S ribosomal RNA was used as an internal control standard. Relative expression levels were analyzed using the comparative 2^−ΔΔCt^ method.

For northern blotting, 3–5 μg of total RNA was denatured in 50% formamide at 68 °C for 3–5 min, separated on agarose gels with 1% formaldehyde in MOPS buffer (pH 7.0), and transferred onto nylon membranes. Digoxigenin (DIG)-labeled DNA probes specific for the 3′-untranslated region of the CLBV genome (nucleotides 8315–8626) and CLBV MP (nucleotides 544–902) were used and prepared using a PCR DIG Probe Synthesis Kit (Roche, Basel, Switzerland). The hybridization conditions and detection of RNAs were carried out as described in the DIG Application Manual supplied by Roche.

## 3. Results

### 3.1. CLBV Infection Activates Autophagy

To understand the interplay between autophagy and CLBV, we examined the autophagy activity following CLBV infection. An infectious cDNA clone of CLBV based on a binary vector plasmid was generated and used to inoculate *N. benthamiana* by agroinfiltration. At 14 days post-inoculation (dpi), the upper leaves of inoculated plants showed leaf curl with yellowing or chlorotic symptoms (Figure 1A) and virus infection was detected in non-inoculated upper leaves by Western blot analysis using CLBV CP antiserum (Figure 1B). ATG8 is conjugated to phosphatidylethanolamine (PE) lipids during the formation of autophagosomes. As the level of accumulation of lipidated ATG8 (ATG8-PE) indicates the level of autophagic activities in the cell [59,60], we examined the accumulation of modified ATG8 following CLBV infection by immunoblotting using an *Arabidopsis* ATG8a antibody, which was previously used to detect ATG8 accumulation in various plant species [61,62,63]. SDS-PAGE in the presence of urea was carried out to separate ATG8-PE from ATG8 [64]. Immunoblotting detected unmodified ATG8 and ATG8-PE as slower and faster-migrating bands, respectively (Figure 1B). Upon CLBV infection, accumulation of both ATG8 and ATG8-PE was elevated (Figure 1B), suggesting that autophagy was activated during virus infection. Next, we examined the ultrastructure of autophagic bodies in the vacuoles by TEM observation. Much more numbers of autophagic body-like structures were observed in the vacuole of cells of CLBV-infected plants than in non-infected plants (Figure 1C, arrows). Quantification of the structures showed that the number of autophagic bodies in CLBV-infected plants was substantially increased (by approximately 5.0 fold) compared to that in non-infected plants (Figure 1D). Fluorescent protein-tagged ATG8 has been widely used to monitor autophagic activity [65]. We transiently expressed green fluorescent protein-tagged *N. benthamiana* ATG8f (GFP-NbATG8f) in leaf tissues and observed the labeled autophagic bodies in CLBV-infected and non-infected plants by fluorescence microscopy (Figure 1E). Consistent with the TEM observation of autophagic bodies, CLBV infection induced an approximately 3.0-fold greater number of GFP-NbATG8 punctate fluorescent signals relative to non-infected plants (Figure 1F). Collectively, these data indicate that autophagy was activated during CLBV infection.

### 3.2. Autophagy Inhibits Systemic Infection of CLBV

Since autophagy is activated by CLBV infection (Figure 1), we next investigated its antiviral role. To impair the autophagy system, the autophagy-related genes *N. benthamiana* (*Nb*) *ATG5* and *NbATG7* were silenced using TRV-VIGS [55,65]. At 10 days after inoculation with TRV vector virus, when the levels of *NbATG5* and *NbATG7* mRNAs were significantly reduced in TRV-NbATG5 and TRV-NbATG7-infected plants as compared to non-silenced control plants (TRV-GFP-infected plants) (Appendix A), the plants were inoculated with a CLBV infectious clone. At 10 dpi, a mild chlorotic symptom appeared on the newly emerged upper leaves of *NbATG5*- and *NbATG7*-silenced plants, whereas there were no visible symptoms on control plants (TRV-GFP infected) (Figure 2A). Accordingly, the accumulation levels of CLBV CP and genome RNA assessed by Western blot and qRT–PCR, respectively, in *NbATG5*- and *NbATG7*-silenced plants were markedly higher than those in control plants (Figure 2B,C). At 14 dpi, the upper leaves of control plants showed mild chlorotic symptoms similar to those of *NbATG5*- and *NbATG7*-silenced plants (Figure 2A). At this point, the accumulation levels of CLBV CP and genome RNA were slightly higher in *NbATG5*- or *NbATG7*-silenced plants than in control plants but not significantly different according to statistical analysis (Figure 2B,C). These observations suggested that autophagy specifically interfered with the progress of CLBV systemic movement but did not affect CLBV accumulation when the systemic infection was established in the upper leaves. To gain a more detailed view of the effect of autophagy on CLBV systemic movement, an inoculation experiment was carried out in which virus RNA accumulation was monitored in the newly emerged leaves at 5, 7, 10, and 14 dpi by Northern blot analysis. In the *NbATG5*- and *NbATG7*-silenced plants, CLBV RNAs were readily detectable at 7 dpi, while in control plants, CLBV RNAs were first detected at 14 dpi (Figure 2D). Moreover, in control plants, RT-PCR detected a low level of CLBV RNA accumulation at 10 dpi but not at 7 dpi (Figure 2D). Consistently, in a separated inoculation experiment, at 7 dpi CLBV symptoms and genome accumulation were detected in the upper leaves of *NbATG5*- and *NbATG7*-silenced plants but not in those of non-silenced control plants (Appendix A). Thus, it is obvious that impairing autophagy largely accelerates the systemic movement of CLBV in *N. benthamiana*.

### 3.3. Autophagy Inhibits Cell-to-Cell Movement of CLBV

Viral spread throughout plants generally consists of two steps: cell-to-cell movement (local spread) through the plasmodesmata and long-distance transport through the vasculature [66,67]. To further investigate whether autophagy affects the cell-to-cell movement of CLBV, we generated a GFP-expressing CLBV variant (CLBV-GFP). A GFP gene was inserted downstream of the CP coding region in the infectious clone of CLBV. A duplicate of the subgenomic promoter sequence of CP was also added so that GFP could be expressed through sgRNA transcription (GFP sgRNA; Figure 3A) [48]. As expected, northern blotting confirmed that the GFP sgRNA was transcribed during CLBV-GFP infection in the upper systemic leaves of *N. benthamiana* inoculated via agroinfiltration (Figure 3B). CLBV-GFP was then inoculated into *NbATG5*- and *NbATG7*-silenced plants. In order to obtain CLBV-GFP infection that initiated from a single cell, an *Agrobacterium* culture harboring the infectious clone of CLBV-GFP was 10,000-fold diluted from an optical density at a wavelength of 600 nm (OD_600_) of 1 and used to infiltrate the leaves. Fluorescence-microscopy observation of the infiltrated leaves showed that at 5 days after infiltration, GFP fluorescence was predominantly restricted to a single cell in the leaves of non-silenced plants, whereas GFP fluorescence was mostly seen as clusters consisting of six or seven cells in leaves of *NbATG5*- and *NbATG7*-silenced plants (Figure 3C). The number of cells in 10 GFP-expressing foci was significantly higher in the leaves of *NbATG5*- and *NbATG7*-silenced plants than in those of non-silenced control plants (Figure 3D). This result suggests that autophagy inhibits the cell-to-cell movement of CLBV and this negatively affects the overall spread of CLBV throughout the plants.

### 3.4. CLBV MP Is Targeted for Autophagic Degradation

To elucidate the mechanism of autophagy-mediated antiviral defenses against CLBV infection, we investigated the CLVB-encoded proteins that were targeted by autophagy. CLBV replicase, MP, and CP were fused with GFP (Replicase-GFP, MP-GFP, and CP-GFP) and transiently expressed in *N. benthamiana* plants. Immunoblotting using a GFP-specific antibody could detect MP-GFP and CP-GFP but not Replicase-GFP although Replicase-GFP transcripts were detected by RT-PCR (Figure 4A and Appendix A). Replicase-GFP may have been expressed below the limit of the detection level. Leaf tissues transiently expressing Replicase-GFP, MP-GFP, and CP-GFP were then treated with the lysosomal protease inhibitor E64d and the class III PI3K inhibitor 3-methyladenine (3-MA), which both inhibit the autophagy pathway, and MG132, a 20S proteasome inhibitor. Immunoblotting showed that accumulation of MP-GFP, but not the two other fusion proteins, was increased upon treatment with autophagic inhibitors (3-MA or E64d), whereas treatment with the protease inhibitor MG132 did not affect the accumulation of all three fusion proteins as compared with the control treatment dimethyl sulfoxide (DSMO) (Figure 4A and Appendix A). This suggests that CLBV MP is subjected to autophagic degradation. To further confirm this observation, HA-tagged MP and CP (MP-HA and CP-HA) were transiently expressed in the leaves of *NbATG5* and *NbATG7*-silenced plants. Consistent with the result of chemical treatments, increased accumulation of MP-HA, but not CP-HA, was observed in *NbATG5*- or *NbATG7*-silenced plants as compared to non-silenced control plants (Figure 4B).

To examine the association of CLBV MP and CP with autophagosomes, MP and CP were fused with mCherry (MP-mCherry and CP-mCherry) and co-expressed with GFP-NbATG8f, which was used as an autophagosome marker [65]. Fluorescence-microscopy observation showed that MP-mCherry was localized in small granule-like or punctate structures, many of which were co-localized with similar structures labeled by GFP-ATG8f. In contrast, CP-GFP was distributed throughout the cytosol and showed no association with the structures labeled by GFP-ATG8f (Figure 4C,D). Collectively, these data suggest that CLBV MP is engulfed by autophagosomes and degraded via a selective autophagy pathway.

### 3.5. CLBV MP Interacts with NbATG8C1 and NbATG8i

Here we investigated whether CLBV MP was directly recognized and recruited by ATG8. The *N. benthamiana* genome encodes 13 ATG8 isoforms, which can be divided into four groups based on phylogenetic relationships (Figure 5A). We examined the interactions between MP and a representative ATG8 isoform from each group, namely NbATG8C1, NbATG8d, NbATG8f, and NbATG8i, using a BiFC assay [68]. These proteins were fused to the *N*-terminal or C-terminal portions of the YFP [YFP(n) or YFP(c), respectively] and then co-expressed in *N. benthamiana*. A BiFC assay showed that MP interacted with NbATG8C1 and NbATG8i, but not with NbATG8d and NbATG8f (Figure 5B). Notably, the reconstituted yellow fluorescence indicating the protein-protein interactions was predominantly localized in small granular-like or punctate structures resembling autophagic bodies (Figure 5B). This observation suggests that CLBV MP is associated with ATG8 in autophagosomes. To further confirm this interaction, we performed an in vitro pull-down assay with prokaryotically-expressed CLBV MP fused to the MBP (MBP-MP) and GFP-NbATG8C1, GFP-NbATG8d, GFP-NbATG8f, and GFP-NbATG8i expressed in plants. Consistent with the result of the BiFC assay, GFP-NbATG8C1 and GFP-NbATG8i were co-purified with MBP-MP, indicating protein interactions (Figure 5C).

### 3.6. N-terminal Region Containing AIM Is Important for Recruitment of CLBV MP to Autophagosomes

Because CLBV MP interacts with NbATG8C1 and NbATG8i (Figure 5), we analyzed whether CLBV MP contains AIM. An online bioinformatics analysis (http://repeat.biol.ucy.ac.cy/iLIR) (accessed on 1 September 2020) [69], predicted a potential AIM (LGFVPI) at amino-acid (aa) positions 85–90 in the *N*-terminal region (Figure 6A). To examine the role of this sequence motif, we generated two GFP-tagged MP mutants that lack the entire AIM sequence (MP_Δ85–89_) and contain four alanine substitutions in the AIM sequence (MP_LGAAAA_). However, neither GFP fluorescence nor protein accumulation of MP_Δ85–89_—GFP and MP_LGAAAA_—GFP was detected, although their mRNAs were normally transcribed (Appendix A). Treatment with MG132 slightly enhanced MP_Δ85–89_-GFP but not MP_LGAAAA_–GFP, while treatment with 3-MA or E64d did not affect the accumulation of these two fusion proteins (Appendix A). These MP mutants are not degraded by autophagic pathways but are likely to be structurally unstable and thus difficult to characterize. To gain insight into whether this AIM sequence has a role in the interaction, the half *N*- and C-terminal parts of MP (MP-N and MP-C, Figure 6A) were tested with respect to their binding to NbATG8C1 and NbATG8i by an MBP pull-down assay. Analysis showed the interaction of both N- and C-terminal regions with NbATG8C1 or NbATG8i (Figure 6B). Moreover, the BiFC assay indicated that both N- and C-terminal regions could interact with NbATG8C1 or NbATG8i (Figure 6C), suggesting that the AIM sequence in the *N*-terminal region was not critical for the interaction between CLBV MP and NbATG8C1 and NbATG8i. Nonetheless, it was observed that the YFP fluorescence reconstituted by the interaction with the *N*-terminal part was predominantly localized in the granular-like and punctate structures, while the fluorescence reconstituted by the interaction with the C-terminal part was dispersed throughout the cytosol (Figure 6C). This observation suggests that although both N- and C-terminal regions of CLBV MP contribute to the binding to NbATG8C1 and NbATG8i, the *N*-terminal region is specifically required for the association of MP with autophagosomes. Furthermore, in the BiFC assay, the short *N*-terminal fragment that retains the AIM sequence (MP–N90, Figure 6A), and NbATG8C1 continued to interact in granular-like and punctate structures, while the interaction of the short *N*-terminal fragment lacking the AIM sequence (MP–N84, Figure 6A) and NbATG8C1 appeared not to occur in such structures and was distributed throughout the cytosol (Figure 6C). In contrast, the interaction of NbATG8i with MP–N90 as well as with MP-N84 was not localized in any particular structures and was distributed throughout the cytosol (Figure 6C). This observation suggests that the predicted AIM sequence is sufficient to mediate the recruitment of CLBV MP to autophagosomes via NbATG8C1, while recruitment via NbATG8i also requires other sequences that reside in the half *N*-terminal region.

## 4. Discussion

During the evolutionary battle between hosts and viruses, the former has evolved various defense mechanisms against infection [70,71]. Among them, autophagy has emerged as an important mechanism in immunity against viruses [19,20,72,73,74]. The antiviral role of autophagy in plants has relatively recently been recognized; therefore, the implications of autophagy on plant virus infection have so far been demonstrated only on a limited number of viruses. In this study, we investigated how autophagy affects CLBV infection in an experimental model host plant, *N. benthamiana*. By impairing autophagy through silencing of *NbATG5* and *NbATG7*, we observed that autophagy inhibited the progress of the cell-to-cell movement and systemic infection but had no obvious effect on CLBV accumulation and symptom expression after systemic infection had been established (Figure 2 and Figure 3). Previous reports showed that silencing of *NbATG5* and *NbATG7* in *N. benthamiana* largely promoted plant virus accumulation and symptom expression such as that observed for RSV [30], cotton leaf curl multan virus rus (CLCuMuV) [17], barley stripe mosaic virus (BSMV) [75], and tomato leaf curl Yunnan virus (TLCYnV) [18]. As autophagy operates through targeting diverse viral proteins that function in particular steps of virus infection and life cycle, conceivably, autophagy could exert different effects among viruses. Our results provide an understanding that although autophagy is an intracellular process, its antiviral activities could manifest as suppression of the intercellular spread of the virus rather than the restriction of virus accumulation in the cells.

Inhibitor treatment assays and silencing of *NbATG5* and *NbATG7* revealed that transiently expressed CLBV MP was selectively targeted for autophagic degradation (Figure 4). To establish an infection throughout the plant, viruses depend on the activity and stability of virally encoded MP. In the case of (+)ssRNA, MPs commonly bind viral genomic RNA and enlarge the size exclusion limits of plasmodesmata, but they are not directly involved in viral genome replication [76]. Thus, the inhibition of intercellular transport of CLBV could be attributed to the degradation of CLBV MP by the autophagy pathways. Plants appear to have evolved various pathways to degrade viral MPs. Previously, the 30K MP of the tobacco mosaic virus was shown to be degraded by the 26S proteasome during viral infection [77]. Likewise, the 69K MP of the TuMV was found to be polyubiquitinated and subsequently degraded by the proteasome [78]. It is still unclear whether degradation of viral MPs by the proteasome pathway is a part of host antiviral responses or is a strategy employed by viruses to facilitate effective infection. There is experimental evidence showing that suppression of ubiquitination results in inhibition of virus infection [79,80,81]; this may suggest that in some cases, MP degradation by the proteasome pathway may be beneficial for the viral invasion of plants.

As CLBV MP was shown to have an RNA-silencing suppression activity [46], we could not rule out the possibility that degradation of CLBV MP also led to an inability of the virus to effectively counteract antiviral RNA silencing. In fact, the viral RNA-silencing suppressor is a common target of autophagy degradation [17,28,30,31]. Many viral MPs also function as an RNA-silencing suppressors. For example, P25 of potato virus X [82], 50 kDa MP of apple chlorotic leaf spot virus [83], 29K MP of TRV [84], P4 MP of barley yellow dwarf virus [85], Pns6, a putative MP of rice ragged stunt virus [85], P1, an MP of rice yellow mottle virus [86], MP of red clover necrotic mosaic virus [87], and MP of potato virus M [88]. Furthermore, there is some evidence showing that cell-to-cell or long-distance movement of viruses also requires the activity of viral RNA-silencing suppressors, aside from viral MP [89,90,91,92]. In the *Agrobacterium* co-infiltration assay, CLBV MP showed a weak RNA-silencing suppressor activity as compared to other well-characterized silencing suppressors [46]. It remains to be determined to what extent the RNA-silencing suppression activity of CLBV MP contributes to facilitating the cell-to-cell or long-distance movement of CLBV.

ATG8-family proteins play a central role in the autophagy pathway by interacting with numerous cargo receptors or adaptors via recognition of AIM [10,11]. Some cargo receptors or adapters have been identified to be involved in the selective degradation of plant viral proteins. NEIGHBOR OF BRCA1 (NBR1) acts as a cargo receptor for targeting HC-Pro of TuMV and P4 of cauliflower mosaic virus to autophagosomes [31,32]. Beclin1 (ATG6) interacts with NIb (replicase) of TuMV and targets NIb to autophagosomes possibly via ATG8a [29]. An uncharacterized protein encoded in *N. benthamiana*, NbP3IP, interacts with p3 of RSV and targets it for autophagic degradation through NbATG8f [30]. However, similar to our observation in the current study, there have been previous reports that ATG8 could directly recognize plant viral proteins and be targeted for autophagic degradation. βC1 of CLCuMuB interacts with NbATG8f and the other three *N. benthamiana* ATG8 isoforms [17], while C1 of TLCYnV interacts with ATG8h [18]. Both CLCuMuB βC1 and TLCYnV C1 contain potential AIM sequences but these sequence motifs are not essential for the interaction with the ATG8 isoforms. A TLCYnV C1 mutant with aa substitutions in AIM sequences maintains the ability to bind ATG8h but is defective in the translocation of the ATG8h-C1 complex from the nucleus to the cytoplasm and the induction of C1 autophagic degradation [17,18]. In our study, mutational and deletion analyses suggest that the AIM sequence in CLBV MP is critical for protein structural stability but seems to be dispensable for the interaction with NbATG8i and NbATG8C1 (Figure 6). However, the AIM sequence in CLBV MP, in part, may be necessary for the recruitment of the CLBV-ATG8 complex to the autophagic bodies (Figure 6). Further detailed studies are necessary to elucidate the precise role of AIM in facilitating the recruitment of viral protein targets to autophagosomes.

## Figures and Tables

**Figure 1 viruses-13-02189-f001:**
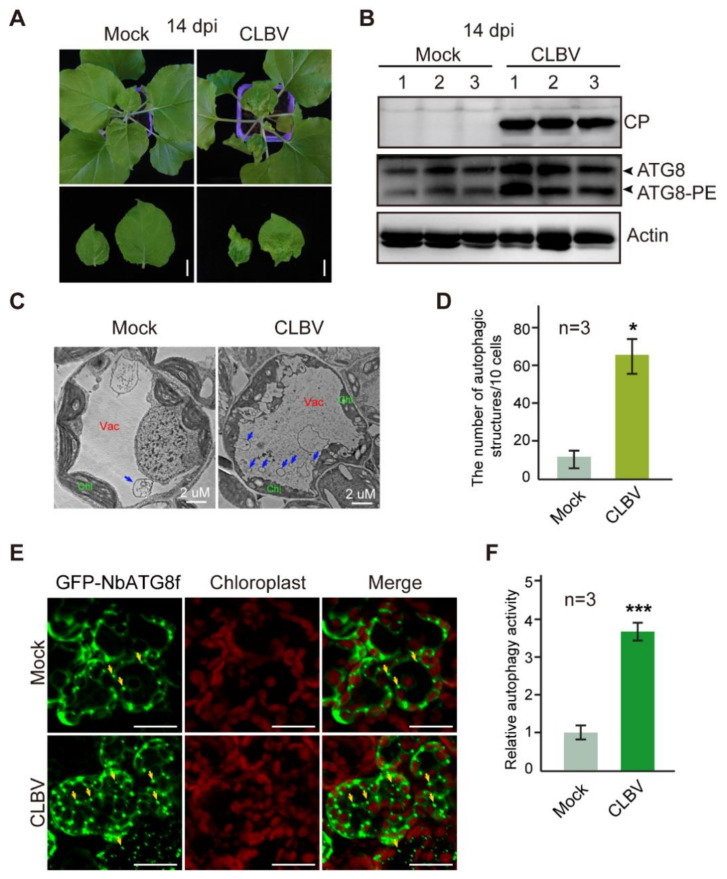
Autophagy activity in CLBV-infected *N. benthamiana*. (**A**) CLBV symptoms in infected plants. Lower panels are images of the upper leaves. Scale bars, 1 cm. (**B**) Immunoblotting analyses of CLBV CP and ATG8 in plants described in (**B**). Total proteins were extracted from upper leaves and used for immunoblotting with a CLBV CP antibody to detect CP and an *Arabidopsis* ATG8a antibody to detect ATG8 and actin antibody to verify equal loading of protein samples. (**C**) Representative TEM images of autophagic body-like structures (arrows) in vacuoles (Vac) of leaves infected with CLBV. Chl, chloroplast. (**D**) Quantification of the numbers of autophagic structures in the cells of leaves described in (**C**). Each bar represents the mean number of structures from 10 cells obtained from three independent experiments. Vertical lines on the bars represent the standard deviation. “*” indicates a significant difference (*p* < 0.05, Student’s *t*-test). (**E**) Autophagic activity following CLBV infection assessed by using an autophagy marker GFP-NbATG8f. The GFP fluorescence in epidermal cells was observed by confocal laser scanning microscopy. Yellow arrows indicate some autophagosomes in the cytoplasm. Scale bars, 20 μm. (**F**) Quantification of autophagic activity based on the numbers of GFP-NbATG8f-labelled autophagic structures in the cells of leaves described in (**E**). Each bar represents the mean number of structures counted from 100 cells obtained from three independent experiments. The mock sample was set to value 1.0. Vertical lines on the bars represent the standard deviation. “***” indicates a significant difference (*p* < 0.001, Student’s *t*-test).

**Figure 2 viruses-13-02189-f002:**
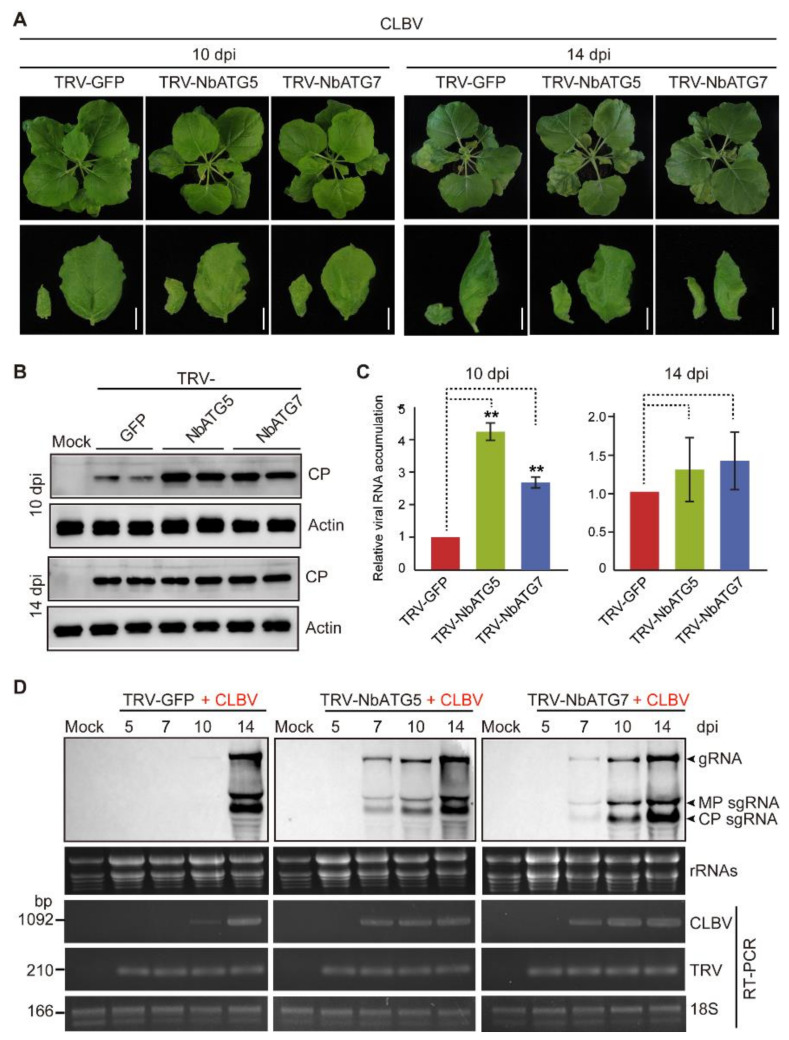
Effect of autophagy on CLBV systemic infection in *N. benthamiana*. (**A**) CLBV symptom expressions in plants with *NbATG5* and *NbATG7* genes had been silenced using TRV-VIGS. Plants were inoculated with TRV–NbATG5 and TRV-NbATG7 or TRV-GFP as a control, and 10 days later, plants were inoculated with CLBV. Lower panels are the images of the upper two leaves. Scale bars, 1 cm. (**B**) CLBV CP accumulation in transgenic plant leaves described in (**A**). Immunoblotting analyses using anti-CP and anti-actin antibodies. (**C**) Quantitative RT-PCR detection of CLBV RNA in the *NbATG5* and *NbATG7* silenced plants described in (**A**). Total RNAs were extracted from the upper leaves at 10 and 14 dpi. qRT-PCR was carried out using primer sets specific for CLBV and *N. benthamiana* 18S rRNA as an internal control standard. TRV–GFP sample was set to value 1.0. The dashed lines indicate two compared samples. “**” indicates a significant difference (*p* < 0.01, Student’s *t*-test). (**D**) CLBV RNA accumulation in infected plant at 5, 7, 10, and 14 dpi. Total RNAs were extracted from upper leaves subjected to RNA blotting analyses with a probe specific for CLBV genome and RT-PCR detection with viral CP gene-specific primers. Ethidium bromide-stained 28S rRNA is shown as a loading control.

**Figure 3 viruses-13-02189-f003:**
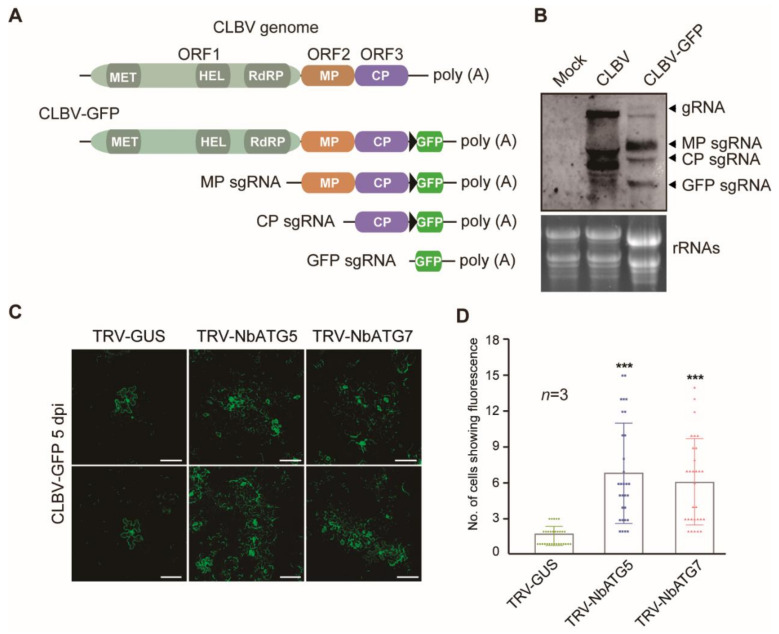
Effect of autophagy on cell-to-cell movement of CLBV in *N. benthamiana*. (**A**) Schematic maps of CLBV genomic and subgenomic RNA with or without GFP insert (not to scale). A black triangle after CP represents a duplicate of the CP subgenomic RNA promoter sequence. MT, methyltransferase motif; Hel, helicase motif; RdRP, RNA-dependent RNA polymerase motif; MP, movement protein; CP, coat protein. (**B**) CLBV and CLBV-GFP RNA accumulation in infected plant. Total RNAs were extracted from upper leaves and subjected to RNA blotting with a probe specific for the CLBV genome. Ethidium bromide-stained 28S rRNA is shown as a loading control. (**C**) GFP expression in epidermal cells of leaves tissue infiltrated with an *Agrobacterium* culture harboring CLBV-GFP after silencing of *NbATG5* or *NbATG7* using TRV-VIGS. At 5 days after infiltration the GFP fluorescence in epidermal cells was observed by confocal laser scanning microscopy. Scale bars, 100 μm. (**D**) Quantification of the numbers of cells in each of the GFP-expressing foci observed in leaves described in (**C**). Each bar represents the mean number of cells from 10 foci obtained from three independent experiments. Vertical lines on the bars represent the standard deviation. “***” indicates a significant difference (*** *p* < 0.001, Student’s *t*-test).

**Figure 4 viruses-13-02189-f004:**
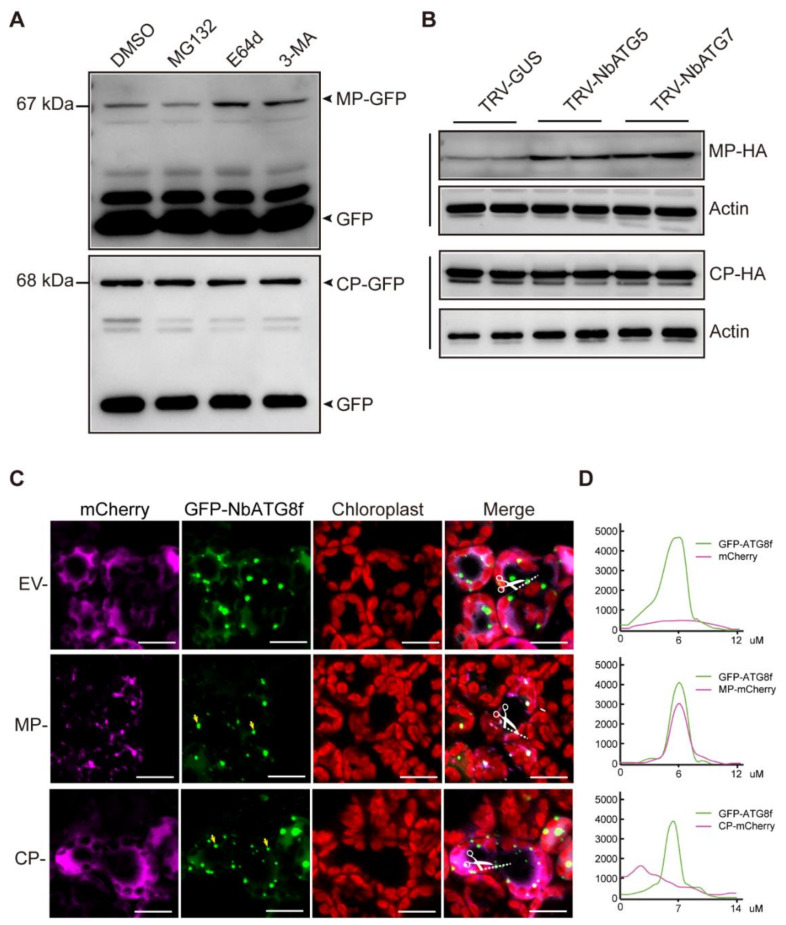
Effect of autophagy on accumulation of CLBV-encoded proteins in *N. benthamiana*. (**A**) Treatment of leaf tissue transiently co-expressing unfused GFP and MP-GFP or CP-GFP with MG132 (20S proteasome inhibitor), and the E64d and 3-MA (autophagy inhibitors). Leaves were infiltrated with inhibitors at 3 days after agroinfiltration and 8 h later the leaves were sampled and analyzed by immunoblotting with a GFP antibody. (**B**) Accumulation of transiently expressed MP-GFP and CP-GFP in the leaves of plants with silencing of the *NbATG5* and *NbATG7* genes mediated by TRV–VIGS. Plants were inoculated with TRV VIGS vectors (TRV-NbATG5 and TRV-NbATG7 or TRV-GUS as a control); 10 days later, leaves were infiltrated with agrobacterium cultures carrying the binary vector construct; and 3 days later, leaves were sampled and analyzed by immunoblotting with anti-GFP and anti-actin antibodies. (**C**) Subcellular localization of MP-GFP and CP-GFP transiently co-expressed with an autophagosome marker GFP-ATG8f. *Agrobacterium* cultures carrying the binary vector constructs were used to infiltrate leaves; 3 days after infiltration leaves were sampled and the GFP fluorescence in epidermal cells was observed by confocal laser scanning microscopy. Yellow arrows indicate some autophagosomes in the cytoplasm. EV, empty vector. Scale bars, 20 μm. (**D**) Histograms of fluorescence intensity of the regions marked by dashed lines in images in (**C**). Fluorescence intensity is shown in arbitrary units.

**Figure 5 viruses-13-02189-f005:**
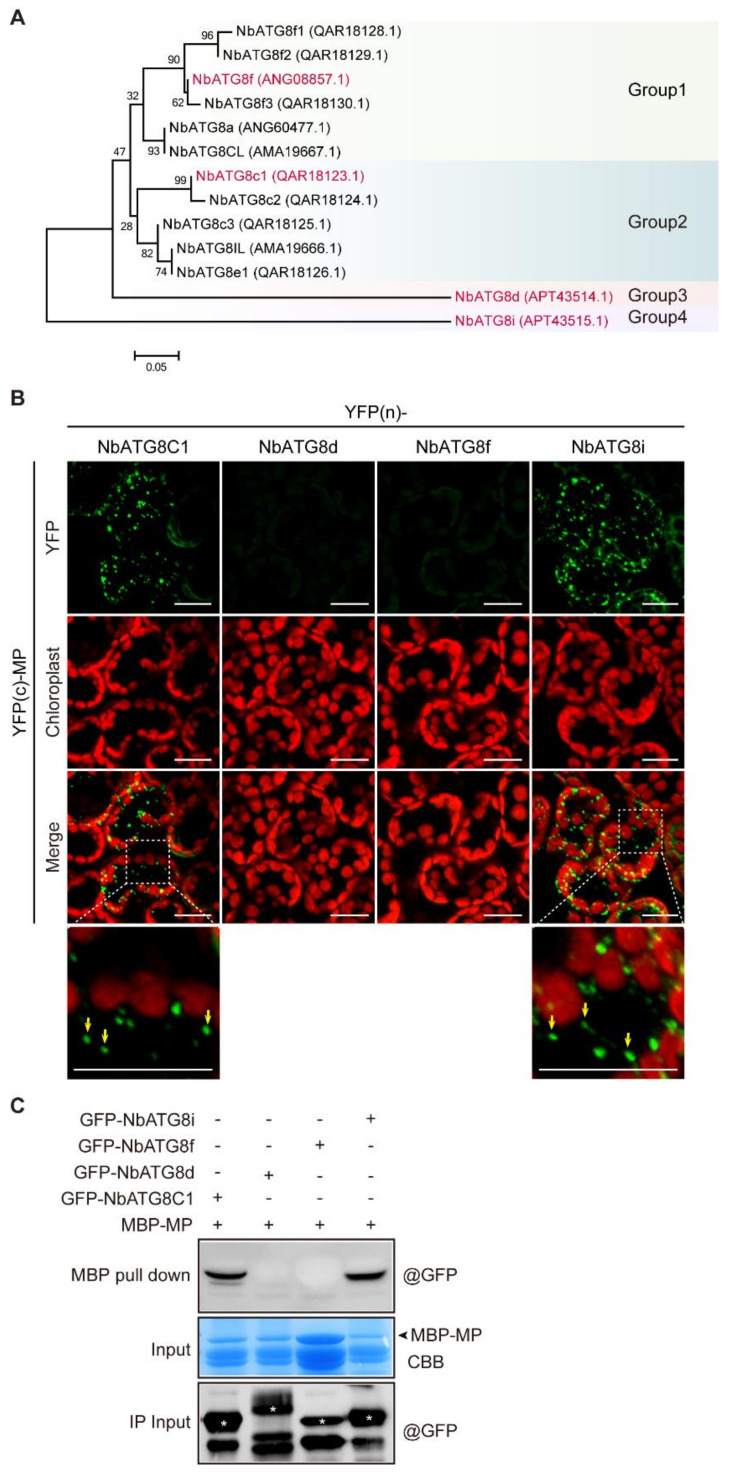
Interaction of CLBV MP and ATG8 isoforms encoded by *N. benthamiana*. (**A**) Phylogenetic analysis of ATG8 isoforms encoded by *N. benthamiana*, *A. thaliana,* and wheat (*Triticum aestivum*). The GenBank accession numbers are shown in brackets. The tree was constructed by the minimum evolution method of Mega 7. (**B**) Interactions of CLBV MP and *N. benthamiana* ATG8 isoforms in BiFC assays. CLBV MP and ATG8 isoforms were fused to *N*-terminal or C-terminal portions of the YFP [YFP(n) or YFP(c)] and then transiently co-expressed using agroinfiltration. Leaves of plants were sampled 4 days after infiltration and the reconstituted YFP fluorescence in epidermal cells was observed by confocal laser scanning microscopy. Enlarged images in rectangle areas are shown in the lower panels. Yellow arrows indicate punctate structures in the cytoplasm. Scale bars, 20 μm. (**C**) Binding of CLBV MP and *N. benthamiana* ATG8 isoforms in an in vitro pull-down assay. Prokaryotically expressed MBP–MP was incubated with GFP-tagged ATG8 isoforms and then MBP pull-downs were performed. Protein samples before and after the MBP pull-downs were subjected to immunoblotting with an anti-GFP antibody. SDS-PAGE of prokaryotically expressed MBP-MP was stained with Coomassie Brilliant Blue (CBB). Asterisks mark GFP-tagged *N. benthamiana* ATG8 isoforms detected with an anti-GFP antibody.

**Figure 6 viruses-13-02189-f006:**
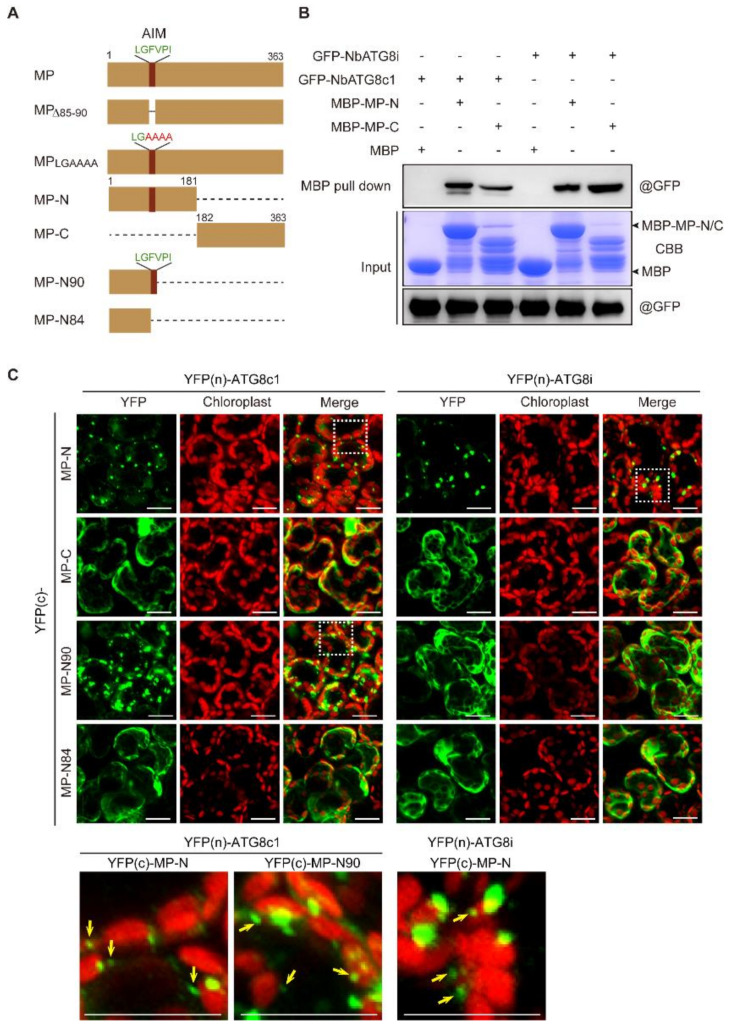
The role of a potential AIM sequence in CLBV MP. (**A**) A schematic diagram of CLBV MP mutants analyzed in this study (not to scale). The position and sequence of AIM is indicated. (**B**) Binding of CLBV MP deletion mutant (MP-N and MP-C) and *N. benthamiana* ATG8 isoforms in an in vitro pull-down assay. Prokaryotically expressed MBP-MP-N and MBP-MP-C was incubated with GFP-tagged NbATG8i or NbATG8c1, and then MBP pull-downs were performed. Protein samples before and after the MBP pull-downs were subjected to immunoblotting with an anti-GFP antibody. SDS–PAGE of prokaryotically expressed MBP-MP-N and MBP-MP-C were stained with Coomassie Brilliant Blue (CBB). (**C**) Interactions of CLBV MP deletion mutants illustrated in (**A**) and NbATG8i or NbATG8c1 in BiFC assays. CLBV MP deletion mutants and ATG8 isoforms were fused to *N*-terminal or C-terminal portions of the YFP [YFP(n) or YFP(c)] and then transiently co-expressed using agroinfiltration. Leaves of plants were sampled 3 days after infiltration and the reconstituted YFP fluorescence in epidermal cells was observed by confocal laser scanning microscopy. Enlarged images in rectangle areas are shown in the lower panels. Yellow arrows indicate punctate structures in the cytoplasm. Scale bars, 20 μm.

## Data Availability

The raw data supporting the conclusions of this article will be made available by the authors, without undue reservation.

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
