# Peer review of "Autophagy Inhibits Intercellular Transport of Citrus Leaf Blotch Virus by Targeting Viral Movement Protein"

_viruses, 2021, doi:10.3390/v13112189_

Round 1
Reviewer 1 Report
General comments:
The manuscript “Autophagy inhibits intercellular transport of citrus leaf blotch virus by targeting viral movement protein” by Niu et al. reports that the infection of citrus leaf blotch virus (CLBV; genus Citrivirus, family Betaflexiviridae) in Nicotiana benthamiana plant activates autophagy mechanism. The autophagy mechanism specifically leads to the degradation of CLBV movement proteins and thus, limiting viral cell-to-cell movement.
In this study, the authors observed autophagy phenomenon in CLBV infected tobacco plants. The autophagy reducing CLBV cell-to-cell and systemic movement was identified when the authors used TRV VIGS system to repair autophagy mechanism. In detected viral proteins, only MP accumulation was increased after autophagy inhibitors treatment. Results of BiFC and pull down assays showed that CLBV MP interacts with two ATG8 isoforms, NbATG8C1 and NbATG8i, of tobacco. The authors also identified the ATG8i interacting domain of CLBV MP. Viral proteins involved in plant autophagy, anti-viral and proviral recognition, were clearly discussed. The research design and methodology were logical, and the results were clear in this report. This study showed a novel example in plant-virus interaction through autophagy mechanisms. I would suggest this manuscript to be accepted for publication in “Viruses”, but some typos and obvious errors must be corrected in order to improve the quality of this manuscript.
Specific comments and suggestion
Page 2, Line 64: The font of period symbol is different to others. This kind of font errors was also found in lines 68 and 386. I would suggest the authors check the entire manuscript carefully to prevent these kinds of errors.
Page 5, Line 215: The word “lapigated” should be “lipidated”.
Page 5, Line 223: “AGT8” should be “ATG8”.
Page 7, Line 264: There is a mistake in the description “the accumulation levels of CLBV CP genome and RNA assessed by western blot and qRT-PCR”. The method of western blot is used to detect the level of CP protein but not CP genome.
Page 10, Line 354, legend of Figure 4A: Fluorescence of Replicase-GFP is not observable that is described in the results section. There is no photo of Replicase-GFP shown in Figure 4, and the term “Replicase-GFP” can be removed from this legend.
Page 15, Lines 468-470 and throughout the manuscript. The name of viruses should following the ITCV rules (https://talk.ictvonline.org/information/w/faq/386/how-to-write-virus-species-and-other-taxa-names). Virus names should not be italicized or capitalized unless they refer to a taxonomic entity.
Author Response
Responses to Reviewer 1:
1. Page 2, Line 64: The font of period symbol is different to others. This kind of font errors was also found in lines 68 and 386. I would suggest the authors check the entire manuscript carefully to prevent these kinds of errors.
Response: We have corrected these errors and checked the entire manuscript carefully. Page 2, lane 59; Page 2, lane 63 etc.
2. Page 5, Line 215: The word “lapigated” should be “lipidated”.
Response: We have corrected the text. Page 5, lane 209
3. Page 5, Line 223: “AGT8” should be “ATG8”.
Response: We have corrected the text. Page 5, lane 216
4. Page 7, Line 264: There is a mistake in the description “the accumulation levels of CLBV CP genome and RNA assessed by western blot and qRT-PCR”. The method of western blot is used to detect the level of CP protein but not CP genome.
Response: We have corrected the sentence. Page 7, lane 258
5. Page 10, Line 354, legend of Figure 4A: Fluorescence of Replicase-GFP is not observable that is described in the results section. There is no photo of Replicase-GFP shown in Figure 4, and the term “Replicase-GFP” can be removed from this legend.
Response: We have removed the texts. Page 11, lane 351
6. Page 15, Lines 468-470 and throughout the manuscript. The name of viruses should following the ITCV rules (https://talk.ictvonline.org/information/w/faq/386/how-to-write-virus-species-and-other-taxa-names). Virus names should not be italicized or capitalized unless they refer to a taxonomic entity.
Response: We have made correction according to ICTV rules. Page 16, lane 464-466

Reviewer 2 Report
This paper describes “Autophagy inhibits intercellular transport of Citrus leaf blotch citrivirus, a Betaflexiviridae family, by targeting viral movement protein” using Nicotiana benthamiana (Nb) and Tobacco rattle virus (TRV) vector which carried autophagy related Nb genes, NbATG5 and NbATG7for VIGS and NbATG8 isoforms for binding assay. The theme and the content are timely and attractive and have novelty for this Journal. However, the manuscript is not well written overall including the references. There are several points to clarify, some flaws, etc. For example, there is no description on their results of TEM examination (Table1C). It needs to describe what is different between mock and virus infected cells in detail otherwise readers cannot understand. No description on vacuole and autophagic bodies in both types of cells. Describe which is vacuole or autophagic bodies by using such as arrowheads (Fig. 1C and 1E). It appears that yellow spots seems to be located on the chlorophyll membranes (Figs. 1E and 4E), but no explanations on those there. It is very strange that no detection of CLBV RNA bands at 7 and 10 days in TRV-GFP (Figs. 2D and S2) but similar or higher level of CLBV RNA band at 14 days (Figs. 2D). There is no explanation on what happened to CLBV during 10 to 14 days. Additional data of TRV RNA accumulation levels might be of some help on this point. From these and other points shown below the manuscript needs vigorous improvements. Thus it is not acceptable in the present form for this Journal. Some comments are described below.
- As a general, reference citation must be described as number in order of appearance in the text instead of author and year. It needs to follow the guideline of this paper.
- Line 32: add reference(s) after mega-autophagy.
- Line 270: Fig. 2C; The expression levels do not appear similar, but different between TRV-GFP and other two at 10-dpi as shown asterisks.
- Line 271: It is not so clear that the systemic movement of CLBV (cell to cell movement or long distance movement?) is interfered. Some more detailed explanation is needed using any other data or evidence if any.
- Line 277: Fig. 2D; As mentioned above, some explanation is needed for no RNA bands at 7 and 10 dpi.
- Line 320: Fig. 3 legend; “in front of” may be replaced by “after”.
- Lines 555-738: References must be numbered in order of appearance in the text (including table captions and figure legends) and listed individually at the end of the manuscript as mentioned in the guideline. Journal name must be abbreviated. Capital or lower case letters, italic/non italic in species name needs to check, depending upon individual reference paper.

Author Response
Responses to Reviewer 2:
There are several points to clarify, some flaws, etc. For example, there is no description on their results of TEM examination (Table1C). It needs to describe what is different between mock and virus infected cells in detail otherwise readers cannot understand. No description on vacuole and autophagic bodies in both types of cells. Describe which is vacuole or autophagic bodies by using such as arrowheads (Fig. 1C and 1E).
Response: We have added arrows and texts to indicate vacuoles and autophagic bodies in the figures. (Figure 1C, 1E and 4C)
It appears that yellow spots seems to be located on the chlorophyll membranes (Figs. 1E and 4E), but no explanations on those there.
Response: Yellow spots (merge of green and yellow colors) around the chloroplast membranes due to overly strong expose of chloroplast autofluorescence (red signal). We have reduced the red signal to minimize the appearance of yellow spots (Figure 1E).
It is very strange that no detection of CLBV RNA bands at 7 and 10 days in TRV-GFP (Figs. 2D and S2) but similar or higher level of CLBV RNA band at 14 days (Figs. 2D). There is no explanation on what happened to CLBV during 10 to 14 days. Additional data of TRV RNA accumulation levels might be of some help on this point.
Response: We have added the results RT-PCR detection of CLBV RNA at 5, 7, 10 and 14 dpi (Figure 2D) and added corresponding texts in the results section (Page 7, lane 274-275) and legend (Page 8, lane 291-292).
1. As a general, reference citation must be described as number in order of appearance in the text instead of author and year. It needs to follow the guideline of this paper.
Response: We have revised the format of references according to referencing style of this journal.
2. Line 32: add reference(s) after mega-autophagy.
Response: We have added the references. Page 1, lane 31
3. Line 270: Fig. 2C; The expression levels do not appear similar, but different between TRV-GFP and other two at 10-dpi as shown asterisks.
Response: We assume that you mean CLBV RNA accumulation at 14 dpi. CLBV genome RNA were slightly higher in NbATG5- or NbATG7-silenced plants than in control plants but not significantly different according to statistical analysis. For better clarity, we have modified the sentence. Page 7, lane 262-265
4. Line 271: It is not so clear that the systemic movement of CLBV (cell to cell movement or long distance movement?) is interfered. Some more detailed explanation is needed using any other data or evidence if any.
Response: This question is addressed in the experiment described in Figure 3.
5. Line 277: Fig. 2D; As mentioned above, some explanation is needed for no RNA bands at 7 and 10 dpi.
Response: As mentioned above, we have added the result of RT-PCR detection to Figure 2D.
6. Line 320: Fig. 3 legend; “in front of” may be replaced by “after”.
Response: We have corrected the texts. Page 9, lane 317
7. Lines 555-738: References must be numbered in order of appearance in the text (including table captions and figure legends) and listed individually at the end of the manuscript as mentioned in the guideline. Journal name must be abbreviated. Capital or lower case letters, italic/non italic in species name needs to check, depending upon individual reference paper.
Response: We have revised the format of references carefully according to the referencing style of this journal.

Reviewer 3 Report
This manuscript describes the role of autophagy in CLBV N. benthamiana infection and the interaction of plant autophagy related proteins with virus proteins. The experimentation are well described and illustrated. I have a general remark concerning some misplacing information, as some background information appears in results and not in introduction, whereas synthesis of the results of the work appear in introduction in place of discussion/conclusion (see attached file). Note that there is an error in figure 6 C.

Author Response
Responses to Reviewer 3:
Detailed modification as follows:
(1) Line 30, we deleted “starvation”. Page 1, lane 29
(2) Line 40, we changed “a” to “an”. Page 1, lane 38
(3) The main results described in last paragraph of the introduction have been shortened into one sentence. (Page 2, lane 83-84). Some information in result section including their references have been moved to introduction section according to your suggestions (Page 2, lane 50-51 and Page 2, lane 69-70). Information regarding virus spread in results section 3.3. (Page 8, lane 294-296) is retained in result section because this information helps the reader to understand the objective of the experiment.
(4) We have exchanged the position between the image of “YFP” and “Merge” of MP-N90 sample (Figure 6C).

Round 2
Reviewer 2 Report
This is the revised manuscript which showed substantial improvements after incorporating the comments by this reviewer. Most of them seem to be appropriate but some more modification and corrections are needed in the text including references. From this point of view the manuscript could be is acceptable after those. They are shown below.
Comments:
Line 70: Nicotiana should be in italic.
Lines 235 and 236: Vac, Chr and Arrows are difficult to see because of the same type of grey color. These should be in highly contrasted colors such as yellow, white, light pink color, respectively.
Lines 241-242 and 360-361: It is difficult to see purple arrows. These should be in higher contrasted color such as light yellow.
Line 289: Two asterisks are confusing here. These two bar graphs are significantly different from control without asterisks or these two are not different (same) between TRV-NbATG7 and TRV-NbATG7? Compared to those two at 14 dpi, these two appear significantly different based on the error bars. It needs to clarify.
Line 384: In Fig 5B and Fig. 6C it is very difficult to see “granular like or punctate structures” there. Add here or as supplementary Figure one or more “enlarged” typical picture(s) or part of that with arrows in high contrasted color.
Line 435: N85 may be N84 according to Fig A?
Line 578: “Solanaceous” should be “solanaceous”. It is dependent on the reference paper.
Line 595: Bamboo mosaic virus should be in italic.
Lines 614, 678, 682, 690 and 710: The Plant Cell should be Plant Cell.
Line 545: “Agrobacterium” should be “agrobacterium”.
Line 641: “Agrobacterium” should be in italic.
Line 617, 619, 621, 623, 629, 633, 643 and 648: Citrus leaf blotch virus should be in italic.
Line 671and 678: Arabidopsis should be in italic.
Line 723: Potato virus X should be in italic.
Line 715: “in vitro” and “Potato virus X” should be in italic.
Lines 729-730; Tobacco rattle virus should be in italic.
Line 735: Rice yellow mottle virus should be in italic.
Line 736: Red clover necrotic mosaic virus X should be in italic
Line 743: Potato Potexvirus X should be in italic
Author Response
Responses to Reviewer 2 (Round 2):
We greatly appreciate your valuable and very helpful comments and suggestions for revising and improving our paper.
Line 70: Nicotiana should be in italic.
---- “Nicotiana” have been italicized. Page 2, line 70
Lines 235 and 236: Vac, Chl and Arrows are difficult to see because of the same type of grey color. These should be in highly contrasted colors such as yellow, white, light pink color, respectively.
---- The color of texts and arrows have been modified as suggested (Figure 1C).
Lines 241-242 and 360-361: It is difficult to see purple arrows. These should be in higher contrasted color such as light yellow.
---- The color of arrows have been modified as suggested (Figure 1E and Figure 4C).
Line 289: Two asterisks are confusing here. These two bar graphs are significantly different from control without asterisks or these two are not different (same) between TRV-NbATG7 and TRV-NbATG7? Compared to those two at 14 dpi, these two appear significantly different based on the error bars. It needs to clarify.
---- The asterisk indicates a significant difference of viral RNA accumulation between control TRV-GFP and TRV-NbATG5 or TRV-NbATG7 at 10 dpi. However, there is no significant difference at 14 dpi (Fig 2C). For clarity, we added lines between the compared samples (Figure 2C).
Line 384: In Fig 5B and Fig. 6C it is very difficult to see “granular like or punctate structures” there. Add here or as supplementary Figure one or more “enlarged” typical picture(s) or part of that with arrows in high contrasted color.
---- The figures have been modified as suggested (Figure 5B and Figure 6C).
Line 435: N85 may be N84 according to Fig A?
---- N85 has been changed to N84. Page 14, line 437
Line 578: “Solanaceous” should be “solanaceous”. It is dependent on the reference paper.
---- “Solanaceous” has been changed to “solanaceous”. Page 18, line 582
Lines 614, 678, 682, 690 and 710: The Plant Cell should be Plant Cell.
---- The texts have been modified. Page 19, line 618; Page 20, line 682; Page 20, line 686; Page 21, line 694; Page 21, line 714
Line 545: “Agrobacterium” should be “agrobacterium”.
---- The text “Agrobacterium” is not present in line 545.
Line 641: “Agrobacterium” should be in italic.
---- The text has been italicized. Page 19, line 645
Line 671and 678: Arabidopsis should be in italic.
---- “Arabidopsis” had been italicized. Page 20, line 675; Page 20, line 681
Line 595: Bamboo mosaic virus should be in italic.
Line 617, 619, 621, 623, 629, 633, 643 and 648: Citrus leaf blotch virus should be in italic.
Line 723: Potato virus X should be in italic.
Line 715: “in vitro” and “Potato virus X” should be in italic.
Lines 729-730; Tobacco rattle virus should be in italic.
Line 735: Rice yellow mottle virus should be in italic.
Line 736: Red clover necrotic mosaic virus X should be in italic
Line 743: Potato Potexvirus X should be in italic
---- According to the International Committee on Taxonomy of Viruses (ICTV), the virus name is not written in italic. (https://talk.ictvonline.org/files/ictv_documents/m/gen_info/7004)
